# Hyaluronic Acid Hydrogels for Controlled Pulmonary Drug Delivery—A Particle Engineering Approach

**DOI:** 10.3390/pharmaceutics13111878

**Published:** 2021-11-05

**Authors:** Dariush Nikjoo, Irès van der Zwaan, Mikael Brülls, Ulrika Tehler, Göran Frenning

**Affiliations:** 1Department of Pharmaceutical Biosciences, Uppsala University, P.O. Box 591, 75124 Uppsala, Sweden; ires.vanderzwaan@farmbio.uu.se; 2Division of Material Science, Department of Engineering Science and Mathematic, Luleå University of Technology, 97187 Luleå, Sweden; 3Early Product Development & Manufacturing, Pharmaceutical Sciences, R&D, AstraZeneca, 43183 Gothenburg, Sweden; mikael.brulls@astrazeneca.com; 4Advanced Drug Delivery, Pharmaceutical Sciences, R&D, Astra Zeneca, 43183 Gothenburg, Sweden; ulrika.tehler@astrazeneca.com

**Keywords:** hyaluronic acid, salbutamol sulphate, spray-drying, urea, glutaraldehyde, drug delivery

## Abstract

Hydrogels warrant attention as a potential material for use in sustained pulmonary drug delivery due to their swelling and mucoadhesive features. Herein, hyaluronic acid (HA) is considered a promising material due to its therapeutic potential, the effect on lung inflammation, and possible utility as an excipient or drug carrier. In this study, the feasibility of using HA hydrogels (without a model drug) to engineer inhalation powders for controlled pulmonary drug delivery was assessed. A combination of chemical crosslinking and spray-drying was proposed as a novel methodology for the preparation of inhalation powders. Different crosslinkers (urea; UR and glutaraldehyde; GA) were exploited in the hydrogel formulation and the obtained powders were subjected to extensive characterization. Compositional analysis of the powders indicated a crosslinked structure of the hydrogels with sufficient thermal stability to withstand spray drying. The obtained microparticles presented a spherical shape with mean diameter particle sizes from 2.3 ± 1.1 to 3.2 ± 2.9 μm. Microparticles formed from HA crosslinked with GA exhibited a reasonable aerosolization performance (fine particle fraction estimated as 28 ± 2%), whereas lower values were obtained for the UR-based formulation. Likewise, swelling and stability in water were larger for GA than for UR, for which the results were very similar to those obtained for native (not crosslinked) HA. In conclusion, microparticles could successfully be produced from crosslinked HA, and the ones crosslinked by GA exhibited superior performance in terms of aerosolization and swelling.

## 1. Introduction

The pulmonary route has attracted considerable attention in recent decades, due to the capability of the lungs to absorb pharmaceuticals for either systemic or local delivery [1,2,3,4]. The pulmonary route affords a noninvasive administration for the delivery of therapeutic agents due to the considerable permeability, the large surface area (around 70–140 m^2^ in adults), and the good blood supply of the lungs [4]. Further advantages such as the avoidance of first-pass metabolism and the low enzymatic activity result in rapid absorption of therapeutic agents [5].

Drug delivery to the lung is commonly accomplished through an inhaler such as a pressurized metered-dose inhaler, a nebulizer, or a dry powder inhaler. The dry powder inhalers offer some significant advantages in comparison to other technologies/devices (e.g., nebulizers and metered dose inhalers), such as increased stability of the drug substance during storage, operation without the need of a propellant, reduced need to coordinate firing, and inhalation, and potential to deliver high doses [6]. However, most of the marketed inhalation powders are short-acting, required to be inhaled several times a day, resulting in diminishing patient compliance. Sustained release to the lung would be highly beneficial, reducing the daily number of doses, the amount of active ingredient in each dose, systemic side effects, as well as increasing the duration of action and patient compliance. Nonetheless, sustained release to the lung is challenging, as the respiratory tract provides substantial barriers such as clearance by the mucociliary escalator and alveolar macrophages in addition to metabolic degradation (although the metabolic activity of the lung generally is much lower than that of the hepatic system). Thus, there are no sustained-release formulations available in the market [1]. A few clinical formulations have been accomplished to evaluate the potential use of liposomes for sustained pulmonary administration [2], while there is a huge gap between in vitro experiments and clinical investigations for other excipients used, e.g., to formulate polymeric nano/microparticles for pulmonary sustained release [1,7,8,9,10]. Hydrogels encompassing three-dimensional networks have recently attracted much attention in sustained pulmonary delivery, compared to other microparticles, due to their swelling and mucoadhesive features, contributing to bypass barriers for pulmonary drug delivery. The size of swellable microparticles like hydrogels, increases significantly after deposition, which results in mitigation of macrophage clearance [7]. For example, an increase in the size of swellable chitosan-based microparticles has been reported in PBS, where the size of different dry formulations increased from 3–3.5 μm to 30.2–38.1 and 82.7–91.0 μm within 6 and 20 min, respectively [11]. Different natural polymers such as hyaluronic acid (HA), chitosan (CS), gelatin, and carrageenan and also synthetic polymers like poly(vinyl alcohol), oligo(lactic acid), poly(lactic acid), and polymers formed from acrylic acid derivatives have been exploited as excipients for the formulation of pulmonary drug delivery systems [3]. Among different polymers, HA has been frequently reported to have a promising performance, considering its biocompatibility, biodegradability, and non-immunogenicity, which is a requirement for all pulmonary drug delivery systems [12,13,14]. HA is an abundant non-sulfated glycosaminoglycan in the extracellular matrices, composed of the repeating disaccharides units of *N*-acetyl-d-glucosamine and d-glucuronic acid, connected by alternating β-1,4 and β-1,3 glycosidic bonds. The use of HA is widespread in different applications like tissue engineering, viscosupplementation, and drug delivery [15,16,17,18]. Recently, HA has been used as an excipient to control drug dissolution and reach sustained respiratory delivery [1]. It affects the inflammatory mediators in the lung as well as agglutination of alveolar macrophages, resulting in a delay of phagocytosis, prolonged lung retention and consequently sustained absorption to the systemic circulation [1,3]. Numerous in vitro and in vivo studies have been carried out to investigate the possibility of using HA in controlled drug delivery systems for the pulmonary route [19]. For example, in vitro and in vivo studies in rats indicated that a hyaluronate (2140 kDa) solution (0.1% *w*/*v*) increased the pharmacological availability and absorption of insulin at pH 7.0, in comparison to an aqueous solution of insulin [20]. In another study, the adsorption and interaction of hydrophilic HA onto the surface of hydrophobic fluticasone propionate particles was investigated, leading to the extension of adsorption with different HA conformations in solution [21]. Moreover, dry powders of HA presented prolonged insulin release in the lungs of beagle dogs, according to the in vivo study. However, a high degree of inter-subject variability was reported for cohesive HA powders due to difficulties with the administration system. In addition, in vitro measurement failed to significantly differentiate between the tested formulations [22]. The release of budesonide, as a poorly water-soluble drug from a HA-based formulation, indicated a prolonged in vivo pharmacological effect of active substance after endotracheal administration in rats. However, the in vitro release of budesonide was not significantly sustained by the HA. The difference was explained by the fact that the used in vitro dissolution method could not mimic the in vivo lung lining fluid conditions [23].

There are various ways to formulate active pharmaceutical ingredients (APIs) into inhalation powders. In addition to the ubiquitous micronization [24,25], spray drying is widely regarded to be a promising method [26]. It is important to control the particle properties of spray-dried powders such as size, morphology, the composition of an aerosol, and surface energy during production, storage and administration [27].

In this study, two different crosslinkers (urea and glutaraldehyde) were used to prepare HA hydrogels. Successful crosslinking was verified by vibrational spectroscopy. The overall purpose of this work was to assess the feasibility of using spray drying to produce inhalation powders from such HA hydrogels. To this end, microparticles were prepared by spray drying from crosslinked as well as native HA. The obtained microparticles were subjected to extensive characterization, including chemical composition, particle morphology and size analysis, surface zeta potential, swelling, in vitro biodegradation and aerosolization performance.

## 2. Materials and Methods

### 2.1. Materials

Sodium hyaluronate (Mw: 1.5 to 2.2 million Da) was purchased from Acros Organics (Fair Lawn, NJ, USA). Glutaraldehyde (GA) 50% aqueous solution, urea (UR), ethanol (EtOH), 2-propanol and hydrochloride acid (HCl) were purchased from Merck (Darmstadt, Germany). Lysozyme (Muramidase), from hen egg white, was from VWR Chemical (VWR International GmbH, Langenfeld, Germany). Potassium phosphate pH 7.5 was used as a buffer. Deionized water (Milli-Q) was used throughout the sample preparation and purification procedure.

### 2.2. Formulation of Hyaluronic Acid-Based Hydrogels and Hydrogel Microparticles

The HA-based hydrogel microparticles were prepared by chemical crosslinking followed by spray drying. The hydrogels based on HA and two different crosslinkers, namely GA and UR, were formulated and coded as HAGA and HAUR, respectively. For the HAGA hydrogel, the aqueous solutions of HA (1% *w*/*v*) and GA (50%) were mixed in a total volume of 20 mL and stirred to obtain HA-based hydrogels, where the reaction was catalyzed by acid (pH = 2.6) at room temperature for 24 h [28]. The concentration of the crosslinker was adjusted from 1% to 16% *v*/*v* in order to study the effect of different concentrations on the crosslinking procedure. The obtained gels will henceforth be denoted as HAGA1–HAGA16, where the number indicates the amount of crosslinker used. The HAUR hydrogel was prepared as reported by Citernesi et al. [29]. Briefly, the aqueous solutions of HA (1% *w*/*v*) and UR crosslinker were mixed in a total volume of 20 mL. The weight ratio of HA to UR in the aqueous mixtures was changed in the range from 1 to 20 to investigate the effect of different concentrations of UR on crosslinking. The produced hydrogels will henceforth be denoted as HAUR1–HAUR20. The reaction was accelerated by acid (pH = 2.6) at room temperature for 24 h. The obtained hydrogels were mixed with the equivalent amount of deionized water and centrifuged at a relative centrifugal force (rcf) of 2600× *g* (4000 rpm, using a 5430 centrifuge, Eppendorf, Hamburg, Germany) for 30 min. Pure water was used to change the acidic supernatant and the procedure was repeated 3–5 times until the pH was stabilized near neutral (higher than 6). Hydrogel films cast from HAGA1–HAGA16 and HAUR1–HAUR20 were used to assess swelling and the degree of crosslinking (see below). To prepare HA-based hydrogel microparticles, selected purified hydrogel solutions were diluted (0.1% *w*/*v*), using deionized water and stirred for 2 h to reach homogeneous suspensions. The suspensions were fed into a mini spray dryer (B-290, BÜCHI Labortechnik AG, Flawil, Switzerland) with a twin-fluid atomizer (nozzle orifice diameter of 0.7 mm) to obtain solid microparticles. The feed suspensions were kept constantly stirred during spray drying in order to ensure homogenous feeding. The obtained microparticles will be referred to as HAGA and HAUR microparticles for the GA and the UR crosslinker, respectively (Table 1). Spray drying was performed at a feed rate of 0.1 mL/min and an aspiration gas flow of about 30 L/h. By changing the inlet temperature (100, 110, and 120 °C) and atomizing the air-flow rate (357 and 473 L/h), a variety of microparticles with different particle sizes were prepared. The dried microparticles were collected and stored in a desiccator (approx. 1% RH) at room temperature before further analysis. Microparticles based on native (i.e., not crosslinked) HA (denoted HA1-4) were prepared as a reference for the optimization of the spray drying parameters (Table 1). The process conditions used to prepare HA3 were considered optimal and selected for the preparation of hydrogel microparticles. Consequently, HA3 will alternatively be referred to merely as HA microparticles when there is no danger of confusion. The procedure along with the mechanisms for the preparation of hydrogels is schematically illustrated in Figure 1.

### 2.3. Characterization

#### 2.3.1. Swelling of Hydrogel Films

Swelling of the HAGA (thickness 481 ± 58 µm) and HAUR (thickness 360 ± 46 µm) films was evaluated gravimetrically in distilled water at room temperature in order to identify the most suitable degrees of crosslinking for further processing of the gels by spray drying. A height meter (Litematic VL 50A, Mitutoyo, Kawasaki, Japan) was used to determine the film thickness. A dedicated amount of the hydrogels prepared with different crosslinker concentrations was poured into Teflon Petri dishes and dried at 37 °C using an oven (Memmert lab oven, model SM 100, Memmert, Schwabach, Germany). Due to the high viscosity of the hydrogels and, consequently, their nonuniform distribution in the Petri dishes, the obtained dried films were inhomogeneous and brittle. The nonuniform structure of the films may also be attributed to the quick drying of the films using an oven (under air flow and 37 °C) where the elimination of water molecules did not occur at the same rate from the surface and inner parts of the hydrogels. To measure swelling, a determined amount of the film (around 500 mg) was immersed in deionized water. After a certain time, the swollen film was carefully taken out of the water. The excess water was removed using a piece of paper and the gel was then weighed. Water uptake by the HAGA and HAUR films was quantified using the equation:(1)$$\mathrm{Swelling}\;\%=\frac{m-m_{0}}{m_{0}}\times 100$$
where m and $m_{0}$ are the swollen and dry weight of the gel, respectively. Each measurement was performed in triplicate.

#### 2.3.2. Particle Size Analysis

The size distribution of the microparticles was analyzed utilizing a laser diffraction instrument (Coulter LS230, Coulter Corp., Brea, CA, USA). First, 4 mg of microparticles were poured into 4 mL of isopropanol (as an antisolvent) and stirred for 10 min to reach a homogenous dispersion. The dispersion was then sonicated using a water bath (Ultrasonic Cleaner Branson, B5210E-MT, Danbury, CT, USA) for 10 min at a frequency of 47 kHz ± 6%. Thereafter, the well-dispersed specimen was injected into the measuring cell containing isopropanol, equipped with a magnetic stirrer. The particle size analysis was performed using a particle refractive index of 1.66, a dispersant refractive index of 1.37 and an imaginary index of 0.1. Fraunhofer theory was used for the calculation of the particle diameter. The median diameter of a volume distribution and SPAN was determined and reported (averages of three measurements). SPAN was defined as the width of the particle size distribution and expressed by the following equation:(2)$$\mathrm{SPAN}=\frac{d_{90}-d_{10}}{d_{50}}$$
where, $d_{90}$, $d_{50}$, and $d_{10}$ are the 90th, 50th and 10th percentile of the cumulative particle size distribution by volume, respectively.

#### 2.3.3. Zeta Potential

The charging behavior of the microparticles was analyzed using a Malvern Zetasizer Nano-ZS (Malvern Instruments, Worcester-shire, UK) in NaCl electrolyte (10 mM) at a temperature of 25 °C. The electrophoretic mobility was used for the calculation of the ζ potential using Smoluchowski’s equation [31]:(3)$$\zeta =\frac{u_{e}\eta }{\varepsilon_{r}\varepsilon_{0}\;}$$
where ζ represents the zeta potential (mV), η is the dynamic viscosity of the liquid (Pa s), $u_{e}$ represents the electrophoretic mobility (m^2^ V^−1^ s^−1^), while $\varepsilon_{r}$ and $\varepsilon_{0}$ are the relative permittivity of the electrolyte and the electric permittivity of free space (8.854 × 10^−12^ F m^−1^), respectively.

#### 2.3.4. Fourier Transform Infrared (FTIR) Spectroscopy

The chemical structure of the hydrogel microparticles was characterized by Attenuated Total Reflectance Fourier Transform–Infra-Red (ATR-FTIR) spectroscopy. A Bruker VERTEX 80v FTIR spectrometer (Bruker Optik GMBH, Ettlingen, Germany), operating within the range of 500–4000 cm^−1^, with a resolution of 4 cm^−1^ and 128 scans, was utilized.

#### 2.3.5. Scanning Electron Microscopy (SEM)

SEM micrographs of the microparticles were captured using a Leo/Zeiss 1550 (Jena, Germany) microscope, equipped with SmartSEM software. For sample preparation, a suitable amount of specimen was dispersed on the metal stubs by double-adhesive conductive carbon tape. The specimen was thereafter coated by a thin layer of Au/Pd under argon using a sputter coater (Polaron, Quorum Technologies Ltd., Newhaven, UK). Micrographs of microparticles were acquired at different magnifications with an accelerating voltage of 2–3 kV, where surface imaging with high-resolution was performed by an InLens SE detector.

#### 2.3.6. Thermogravimetric Analysis (TGA)

A Q5000 TGA instrument (TA Instruments, New Castle, DE, USA) was used for the thermogravimetric analysis. Approximately 10 mg of microparticles were placed in the center of the platinum sample pan. The sample was heated from 25 °C to 500 °C with a rate of 10 °C/min under Ar gas flow (20 mL/min) and the weight loss was continuously determined. The obtained results were analyzed by TA universal analysis software (version 4.5A, TA instruments).

#### 2.3.7. Differential Scanning Calorimetry (DSC)

A DSC Q1000, (TA instruments) was used for differential scanning calorimetry. First, approximately 2 mg of the microparticles were weighed in an aluminum pan and sealed non-hermetically. An empty aluminum pan was utilized as a reference. After that, the specimens were heated from 25 °C to 500 °C at a constant heating rate of 10 °C/min under an N2 flow of 50 mL/min.

#### 2.3.8. Swelling of Microparticles

For the swelling study, 10 mg of dry microparticles were carefully weighed and placed in an Eppendorf tube containing 1.0 mL of (0.01 M PBS, pH = 7.4). The samples were shaken (80 rpm) using a rotating shaker (Fisherbrand, Multi-Purpose Tube Rotators, Ottawa, ON, Canada) at room temperature. The samples were collected from the shaker and then centrifuged (Eppendorf 5452 MiniSpin Micro Centrifuge, Germany) at 9000× *g* for 2 min at different intervals of (3, 5, 15, 30, 60, and 1440 min). Then, the supernatant was discarded, and the excess water was removed from the tubes with a tiny piece of paper. Finally, the weights of microparticles were recorded and used for the calculation of swelling of microparticles in the same way as for the hydrogel films (cf. Equation (1) above). Each measurement was performed in triplicate.

#### 2.3.9. In-Vitro Biodegradation of Microparticles

The dry microparticles (5 mg), were used for in vitro biodegradation through mixing with 1 mL lysozyme solution (2 mg/mL in 0.01 M PBS, pH = 7.4) in Eppendorf tubes. The tubes were placed in the rotating shaker described above at 80 rpm and room temperature for 1 h to reach swelling equilibrium of the microparticles. Then, the tubes were centrifuged at 9000× *g* as described above for 2 min to separate microparticles from the liquid phase. After that, the supernatant was discarded and excess water was removed from tubes to determine the weight of the swollen microparticles ($m_{0}$) using a balance. The previous steps were repeated at different intervals with swollen microparticles and 1 mL of fresh lysozyme solution. The weights (m) of microparticles were recorded at each interval and enzymatic degradation was calculated by the following equation:(4)$$\mathrm{Remaining}\;\mathrm{weight}\;\%=\frac{m}{m_{0}}\times 100$$
where $m_{0}$ thus is the initial and m the remaining weight of the microparticles at time t, respectively. All samples were tested in triplicate.

#### 2.3.10. Aerodynamic Properties

The aerodynamic properties of the formulations were studied using a modified Andersen Cascade Impactor (mACI) [32]. The modified ACI involves a pre-separator, stage 0, stage 1, and custom-made hollow stages which are used to collect and disperse inhalable formulations onto dedicated filters at ambient temperature. The cutoff diameter of stage 1 is 4.4 μm in this setup and the hollow stages have a height of 2.6 cm similar to the standard stages. For each measurement, 4 mg of each formulation was carefully weighed into a Screenhaler device (AstraZeneca, Goethenburg, Sweden) with a connected Turbuhaler M2 mouthpiece (AstraZeneca, Goethenburg, Sweden). The Screenhaler device was connected to the mACI through the throat. The inhalable formulation was sucked for 0.3 s at 60 L/min through the pre-separator, stages 0 and 1. Then, the formulation was allowed to sediment due to the hollow stages for 20 min onto a filter installed on the filter stage. A modified (i.e., non-compendial) fine particle fraction (mFPF) was calculated based on the weight of the formulation using a balance (METTLER MT5, Mettler Toledo Co., Columbus, OH, USA) with a precision of ± 1 µg. The average and standard deviation from three measurements are reported.

The particle deposition pattern in the mACI was analyzed by SEM. With this aim, the respirable particle dose of formulations was collected on double-adhesive conductive carbon tape mounted on SEM metal stubs located at the filter stage of the mACI. Following the actuation and deposition of the powder formulations, the mACI was disassembled and the SEM stubs with powder deposit were collected. The dispersion of the formulations on the carbon tapes was investigated using SEM as described above.

#### 2.3.11. Water-Sorption Measurement

A Micromeritics ASAP 2020 (Norcross, GA, USA) volumetric gas adsorption analyzer with the “Vapor and Water Vapor Option” was utilized for water vapor adsorption/desorption measurements. All specimens were degassed at 75 °C for 5 h before the measurement using a Micromeritics Smart VacPrep060 (Norcross, GA, USA) sample preparation unit under dynamic vacuum (1 × 10^−4^ Pa). The adsorption/desorption measurements were carried out at isothermal condition (298.15 ± 0.1 K) using a water bath within a vapor pressure range from 0.3 to 2.7 kPa (RH of 10% to 85%).

## 3. Results and Discussion

### 3.1. Swelling and Integrity of Hydrogel Films as Indicators of Crosslinking

The swelling and integrity of the hydrogel films were used as indicators of the extent of crosslinking by GA and UR at different concentrations. Figure 2a shows the effect of GA concentration on the swelling of HA films after 24 h. Coherent gels that exhibited a considerable swelling were formed already for low concentrations of GA. Swelling decreased with increasing crosslinker concentration and reached a plateau when the GA concentration was over 8% or 9% (Figure 2a). An increase in the amount of crosslinker (up to about 8% or 9%) thus resulted in a higher degree of crosslinking and a stiffer gel, as inferred from visual observation and the degree of swelling. There was no significant difference in swelling of the films with a high concentration of GA (>8%), reflecting no significant difference in the crosslinking of the gel. It has been reported that the cytotoxicity of GA as a crosslinking agent is completely dependent on the concentration, i.e., the concentration up to 8% was reported to be non-toxic [33,34]. Still, unreacted GA must be avoided as much as possible; thus, an extensive purification was implemented on synthesized hydrogels. The HAGA8 hydrogel was chosen for the preparation of microparticles using spray drying and further characterizations. For the HAUR films, swelling was measured after 2 h as the films broke down into small parts, making it very difficult to follow the swelling for a longer time. Nevertheless, swelling and integrity of the films were affected by the concentration of UR and more stable films, and consequently larger swelling, was observed for a HA to UR weight ratio of 3:1 (HAUR3 films; Figure 2b). This weight ratio was therefore used for spray drying and further characterization.

### 3.2. Process Conditions and Particle Size Analysis

Particle size is of utmost importance for inhalation powders. The HA hydrogels were spray dried into solid particles with a target volume mean diameter of 1–5 μm and the spray drying parameters were adjusted to reach the desired particle shape and size. According to the results of the particle size measurement by LD (Table 1), the mean diameter of the microparticles varied between 2.3 ± 1.1 and 3.2 ± 2.9 μm, whereas the SPAN ranged between 1.2 and 3.5. The microparticle size decreased with increasing spray gas flow rate from 375 (L/h) to 475 (L/h), and also decreased with increasing inlet temperature from 100 °C to 120 °C (Table 1). An increase in spray gas flow corresponds to a higher supply of energy to break down the liquid into droplets with smaller sizes in the atomization process. The low feed concentration and feeding rate were chosen since higher values for these parameters could result in increased viscosity and larger droplets and consequently larger particles. Normally, an increase in inlet temperature results in larger particles due to the acceleration of liquid evaporation and an early formation of shells [35]. However, no significant effect could be observed (Table 1).

A multimodal particle size distribution was found for the native HA microparticles (HA1-4), corresponding to high SD and span values, while crosslinked HA microparticles (HAGA and HAUR) exhibited a uniform size and unimodal distribution (Figure 3). The multimodal distribution might be caused by powder aggregation due to the higher hygroscopicity of native HA microparticles (see below). Moreover, the structure of the feed suspensions, i.e., the dispersion of native and crosslinked HA, most likely differed. The feed rate of 0.1 mL/min, aspiration gas flow of 30 L/h, inlet temperature 120 °C and atomizing air-flow of 357 L/h were selected as optimized process conditions and the related formulations HA3 (also referred to as merely HA), HAGA, and HAUR (Table 1) were prepared for further characterization. It is worth mentioning that the obtained particle sizes and size distributions are in line with prior results reported for spray-dried HA powder formulations for pulmonary drug delivery [12,23,35,36]. Spray drying of hydrogels is challenging; however, the narrow size distribution (span = 1.2 and 1.3) and mean diameters around 2.2 μm for the crosslinked HA confirm the possibility of using spray drying for preparation of inhalable dry powders based on hydrogels. The results indicate that spray drying not only is convenient for the manufacturing of inhalation powders from HA suspensions but also for the preparation of particles from crosslinked structures.

### 3.3. Zeta Potential

Analysis of the electrophoretic mobility is an efficient method to probe the charge of particles [37]. Zeta potential values are important for understanding and explaining any interaction between different particles in suspension. Zeta potentials of −25.5 ± 8.5 mV, −17.4 ± 4.2 mV, and −18.8 ± 5.4 mV were obtained for the HA, HAGA, and HAUR microparticles, respectively. All microparticles presented a negative charge, while the relatively high values of the zeta potential indicated a stable colloidal system [38].

Furthermore, the surface charge of the microparticles influences the mucoadhesive, mucopenetration, and cell uptake characteristics in mucosal drug delivery. According to a previous study, positively charged polymeric particles represented mucoadhesive properties due to the interaction with negatively charged regions of mucus. Where, the negatively charged particles have proved to enhance mucopenetration through minimizing interaction with the negatively charged parts of mucus, as well as cellular uptake via minimizing interactions with positively charged groups of the lipid membrane [39].

The application of HA as a mucoadhesive biopolymer was reported earlier [40,41]. It was also approved that HA is capable of enhancing the penetration of drugs, comparable to chitosan derivatives [42]. Conclusively, in the mucosal drug delivery system, the surface charge plays a key role in predicting drug uptake capacity even though it is not the only effective parameter.

### 3.4. Fourier Transform Infrared (FTIR)

Chemical crosslinking and spray drying were utilized for the synthesis of hyaluronic acid hydrogels and the preparation of microparticles. FTIR spectroscopy was exploited to investigate the chemical composition of the microparticles formed from native and crosslinked HA (Figure 4). The FTIR spectrum of the native HA microparticles involved a characteristic peak at 2993–3716 cm^−1^ for the –OH and –NH stretching (st) vibration, attributable to the hydroxyl and secondary carboxylic acid amide bonds, respectively. The alkane –CH band (st) appeared at 2920 cm^−1^. The C=O (st) of carbonyl groups was observed at 1600 cm^−1^, symmetric bands (st) of carboxylate salts at 1402 cm^−1^, and the C–O–C (st) symmetric ether bands at 1021 cm^−1^. For the crosslinked HAGA microparticles, the band that appeared at 1600 cm^−1^ in HA, decreased significantly in comparison to that of native HA and shifted to about 1630 cm^−1^ due to the superposition with new bands. The band observed at 1730 cm^−1^ corresponded to unreacted C=O aldehyde groups. In addition, the intensity of the OH peak at 3200 cm^−1^ decreased significantly due to the crosslinking reaction [43,44]. The spectrum of the crosslinked HAUR microparticles exhibited new peaks at 1375 cm^−1^ and 1450 cm^−1^, ascribed to the bending characteristics of –CH. The new peak at around 1650 cm^−1^ is usually attributed to the formation of amide C=O group. Moreover, the intensive peak belonging to the carboxylate salts at 1402 cm^−1^ disappeared, while the –CN (st) peak appeared at 1300 cm^−1^, proving that crosslinking by UR had occurred.

### 3.5. Scanning Electron Microscopy (SEM)

SEM images (Figure 5) of the microparticles indicated that the microparticles were polydisperse. This is a common feature for particles produced by spray drying with a twin-fluid atomizer. The microparticles based on native and crosslinked HA presented a spherical shape and very smooth surfaces with no apparent macroporosity. This morphology is distinct from the one observed in prior studies, in which less smooth particles typically were produced, and may be caused by the absence of additional additives (API or other excipients) or a different structure and rheology of the feed suspension [12,23,35,36,45]. During spray drying, solidification is initiated at the surface of the droplets. Then, the water present in the interior of the formed particles starts to evaporate through vapor diffusion, resulting in voids and pores. However, the lack of macroporosity and the smooth surface indicated that the microparticles continued to condense after solidification at the surface. Indeed, the morphology suggests a continuous phase transition from a dilute aqueous suspension to viscous liquid and solid formation, not in accordance with the theoretical particle formation model [46]. Herein, the microparticles based on the crosslinked HA (Figure 5b,c) exhibited similar morphology to the ones formed from native HA (Figure 5a).

### 3.6. Thermogravimetric Analysis (TGA)

The spray-dried HA, HAGA, and HAUR microparticles were investigated by TGA, revealing their moisture content and the degradation stages of their constituents. The moisture contents for the HA, HAGA, and HAUR microparticles were 8.9%, 6.4%, and 3.9% (*w*/*w*), respectively. A low moisture content (<10% *w*/*w*) has previously been reported for spray-dried microparticles [45,47]. The low moisture content of the formulations facilitates the long-term stability of the inhalation powders. It diminishes the density of the particles and consequently improves their aerosolization propensity [11]. Generally, different factors affect the moisture content of particles prepared by spray drying such as the spray-drying parameters, particle size, and composition. As the spray-drying parameters were the same for all formulations, the difference in moisture content could be referred to differences in particle size distribution and chemical composition. Formulations based on native HA presented wider particle size distributions (Figure 3) than formulations based on crosslinked HA (HAGA and HAUR). Moreover, native HA contains a number of hydrophilic functional groups, such as –OH, –COO and –NH groups, making the structure hydrophilic. GA and UR crosslink via the –OH and –COO groups, respectively, resulting in less hydrophilic structures which are expected to absorb less water.

The thermal analysis (Figure 6) of the HA microparticles presented three stages, consisting of an initial mass loss corresponding to dehydration (20–150 °C), a second stage corresponding to partial breakage of the molecular structure (200–270 °C) and a third stage related to degradation of the hyaluronan residues (270–400 °C) [48,49]. The TGA thermogram of HAUR microparticles presented a dehydration step at 20–150°C with an additional step that can be attributed to the presence of unreacted urea or related chemicals formed from the degradation of crosslinks appearing at 150–223 °C. The two-stage degradation of the polysaccharide was also observed for HAUR microparticles [50]. The moisture evaporation of the HAGA microparticles was observed at 20–150 °C and the degradation of the backbone of the HAGA microparticles started at 200 °C. The obtained results indicate that the microparticle structures in principle could tolerate heating to the inlet temperature used in spray drying (100–120 °C) without a significant impact. Note, however, that the particles typically do not reach such high temperatures, since evaporation of liquid prevents significant heating.

### 3.7. Differential Scanning Calorimetry (DSC)

DSC thermograms of microparticles formed from native (HA) and crosslinked (HAGA and HAUR) HA are shown in Figure 7. The DSC thermogram of HA presented a wide endothermic peak, corresponding to the dehydration process around 108 °C. Polymer degradation was seen as a broad exothermic peak at 240 °C. The obtained results were in accordance with previous reports for HA [51,52,53]. The dehydration peak of HAUR appeared around 85 °C. Furthermore, the DSC thermogram of HAUR exhibited a broad endothermic peak at 185 °C, which might be explained by the presence of unreacted urea or related chemicals formed from the degradation of crosslinks. Furthermore, a wide degradation exothermic peak of the HA polymer was seen around 250 °C [50,52]. The thermogram of HAGA presented a broad endothermic peak around 90 °C, which could be explained by the loss of moisture remaining after the initial drying process. Two exothermic peaks were also observed at higher temperatures (200–250 °C), related to the degradation; the first one represented the conversion of the structure into a less-ordered state and the second peak represented thermal degradation [51].

### 3.8. Swelling of Microparticles

Sustained release in the lung is challenging since suitable microparticles must not only have an adequate size (typically stated as an aerodynamic diameter between 1 and 5 µm) to be able to reach the intended site of delivery but they must also avoid macrophage clearance (which is less effective for particle sizes outside the inhalable range) [54]. Swellable microparticles are therefore promising since they can be tailored to possess respirable aerodynamic sizes in dry form and to obtain larger sizes via swelling upon deposition in the moist environment of the lung [55]. Hence, knowledge of the rate and extent of swelling facilitates the understanding of the biopharmaceutical behavior of the microparticles. Figure 8 presents the gravimetric swelling data for microparticles formed from native (HA) and crosslinked (HAGA and HAUR) HA. All three microparticles exhibited rapid swelling during the first hour. The rapid swelling and consequent increase in the size of the microparticles would diminish their uptake by the phagocytic cells. Indeed, the residence time of the microparticles in the lung is expected to increase with increasing size and weight during swelling. The weight increase after 24 h was larger for the HAGA than for the HAUR and HA microparticles, probably due to the extensive and stable structure formed by crosslinking with GA. The weight increase was consistently somewhat larger for the HAUR than for the HA microparticles. However, overall the swelling was similar for microparticles formed from native HA and from HA crosslinked with UR, and both types of microparticles exhibited a weight loss and decomposition for long periods (6–24 h). These results are consistent with those obtained for the swelling of hydrogel films. The disintegration of the HAUR implies that swelling is not the only parameter with which to measure the stiffness of the hydrogels and other factors such as degree of crosslinking and stability of the crosslinked bonds are of utmost importance.

### 3.9. Biodegradation

The granules of neutrophils and mononuclear phagocytes (e.g., macrophages) in the lung contain the enzyme lysozyme. Lysozyme is also generated by monocytes and epithelial cells [56]. It is therefore important to investigate the effect of this abundant lung enzyme on respirable microparticles. The results obtained from in vitro biodegradation of the microparticle formulations when subjected to lysozyme are displayed in Figure 9. It is noteworthy that the microparticles formed from HA crosslinked with GA (HAGA) degraded slowly with a remaining weight after 24 h of 72% compared to about 30–40% for the other particle types. Similar differences are seen also after 48 h. The reduced biodegradation for HAGA could be explained by the stable and extended network formed by crosslinking with GA, which resulted in a restriction of enzyme diffusion in the network of the hydrogel [45,55]. On the other hand, the biodegradation of the HAUR microparticles was similar to the one for particles formed from native HA. Considering the effects of gel decomposition on swelling, these results can be reconciled with the obtained swelling data for the three formulations. All formulations were biodegraded after 72 h (<10% remaining). For optimal performance in vivo, with administration once daily, an almost complete degradation at 24 h would be desired. In this respect, the HAUR and native HA particles thus seem most attractive. However, the microenvirnment in the lung (e.g., pulmonary surfactants) may also influence the in vivo biodegradation rate. The inclusion of Pluronic F-108 surfactant in a chitin-based formulation has been reported for localized administration in lung solid tumors. Accordingly, the existence of F-108 surfactant in the formulation accelerated the lysozyme-induced biodegradation and consequently the drug release from the microparticles [57].

### 3.10. Aerodynamic Properties

The aerosolization performance of a dry powder formulation is important as only the FPF of aerodynamically classified particles reaches the deeper lung regions after administration [58,59]. Different methods such as the Andersen Cascade Impactor (ACI), the Twin Stage Impinger (TSI), the Next Generation Impactor (NGI), and a special coating chamber connected to the PreciseInhale^®^ system (Inhalation Sciences, Huddinge, Sweden) have been used for collecting FPF for in vitro dissolution studies and further characterization [58]. In this study, a modified ACI (mACI) with a cut-off diameter of 4.4 μm was used and the ratio between the undersized to the total amount of particles was calculated as an indication of the fine particle fraction (mFPF). The thus obtained mFPF was 7.2 ± 3.0% for HA, 9.5 ± 3.2% for HAUR, and 27.8 ± 2.0% for HAGA. Microparticles formed from HA crosslinked with GA (HAGA) thus presented a higher mFPF compared to those for the HA and HAUR microparticles. The obtained mFPF for HAGA is consistent with data reported for spray-dried inhalation powders comprising different polymers, APIs and other additives [12,23,35,36,45,55,60]. However, the mFPF for the HA and HAUR microparticles is lower than typical FPFs reported for the similar formulations. This result could be related to the hygroscopicity and/or strong negative surface charge of the HA-based formulations [12]. Electrostatic charging affects the deposition of inhalation powder in the lungs [61] and adhesion of charged particles to the walls of the inhaler device during loading and discharging has been reported [12]. It is worth noting that the formulations presented in this study do not comprise API and any additives as the aim of the study is to investigate particle engineering of the HA-based hydrogels. Better aerosolization performance could likely be achieved by using standard additives as carrier (i.e., mannitol or lactose) and/or charge modifying agents (i.e., l-lysine or stearylamine) to neutralize the negative charge of the microparticles [12,62].

The deposition patterns of powder formulations collected from mACI were observed by SEM (Figure 10). The HAGA formulation exhibited regularly shaped and well-dispersed microparticles. In contrast, the HAUR and especially the HA formulations presented a few small aggregates comprising small particles located on the surface of the larger particles.

### 3.11. Water-Sorption Measurement

The ability of a material to interact with water vapor provides additional insights into the characteristics of the material, such as its hydrophilicity, hygroscopicity and pore structure. The water vapor sorption data showed that the HA, HAGA, and HAUR microparticles absorbed significant water at high relative humidity (Figure 11) due to the hygroscopic characteristics of the HA-based formulations. Water sorption was reversible, indicating that the water molecules did not chemically bind to the formulations. Consistent with the results obtained by TGA, microparticles formed from native HA (HA) absorbed higher amounts of water (80 mmol/g) than microparticles formed from crosslinked HA (about 50 mmol/g for HAUR and 30 mmol/g for HAGA). As for TGA, the reduced amount of absorbed water can be attributed to the reduction of hydrophilic groups caused by chemical crosslinking by UR and, in particular, GA [63].

The isotherms displayed in Figure 11 concur with Type IV isotherms, indicating a multilayer mechanism of water sorption and weak interaction between adsorbent and adsorbate [64,65]. The hysteresis loops (i.e., the difference of adsorption/desorption isotherms) of the microparticles are consistent with type H3 loops, as obtained for mesoporous structures with slit-shaped pores [65,66]. Note, however, that the interpretation is complicated by the fact that the HA swells in contact with water.

## 4. Conclusions

This study addressed the feasibility of using spray drying to prepare inhalation powders for sustained pulmonary drug delivery from cross-linked hyaluronic acid. To this end, spray drying was used to prepare microparticles from native HA and HA crosslinked with urea and glutaraldehyde and the obtained powders were subjected to extensive characterization. Compositional analysis of the formulations indicated a crosslinked structure of the hydrogels with sufficient thermal stability to withstand spray drying. Microparticles formed from crosslinked HA had a spherical particle shape and mean diameter particle sizes from 2.3 ± 1.1 μm to 3.2 ± 2.8 μm, indicating that they would be suitable for inhalation. However, only the particles crosslinked with glutaraldehyde exhibited adequate aerosolization performance. These particles also showed the most favorable swelling characteristics. It could nevertheless be concluded that spray drying can be utilized to formulate inhalation powders of hydrogels as promising scaffolds for pulmonary sustained drug delivery. Further efforts will be focused on the investigation of drug loading/delivery properties of the inhalation powders for use in sustained pulmonary drug delivery.

## Figures and Tables

**Figure 1 pharmaceutics-13-01878-f001:**
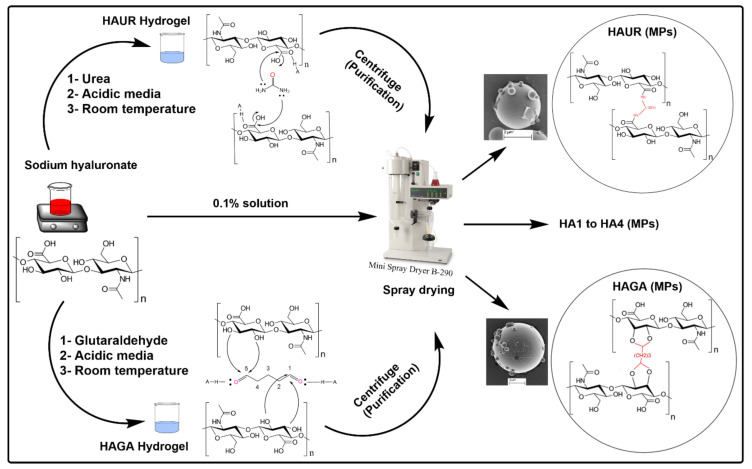
Schematic representation of the preparation of HA-based microparticles (MPs) along with the chemical reaction mechanisms for crosslinking with urea (UR) and glutaraldehyde (GA).

**Figure 2 pharmaceutics-13-01878-f002:**
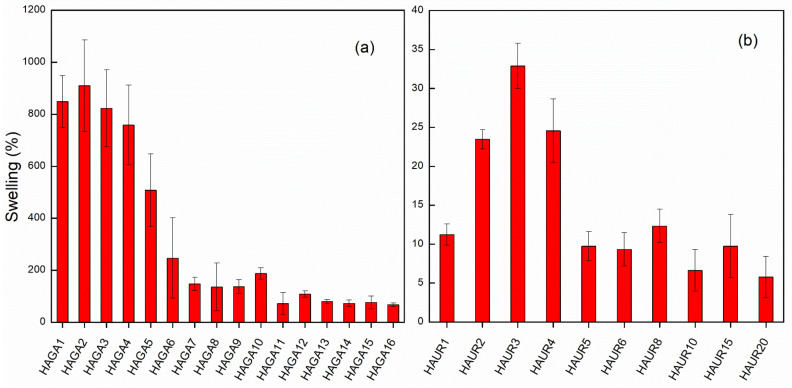
Swelling in deionized water of HA films crosslinked with GA after 24 h (**a**) and UR after 2 h (**b**) with various crosslinker concentrations. Bars represent the standard error of the mean.

**Figure 3 pharmaceutics-13-01878-f003:**
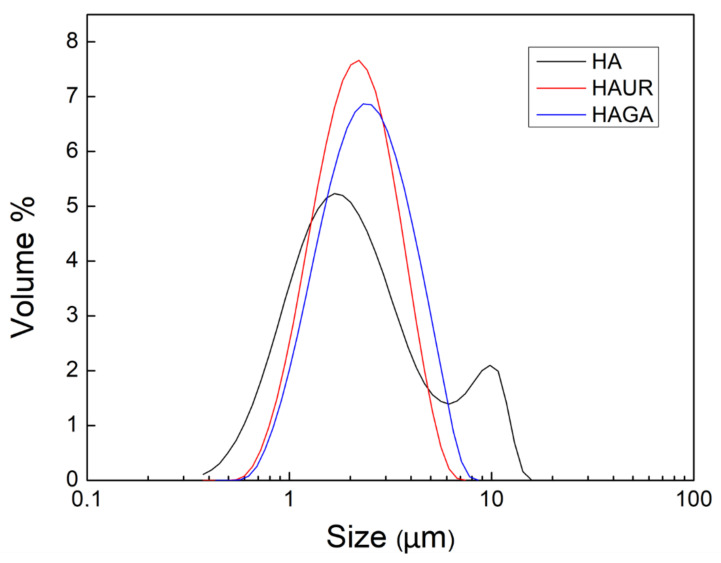
Size distribution of microparticles formed from native (HA) and crosslinked (HAUR and HAGA) HA.

**Figure 4 pharmaceutics-13-01878-f004:**
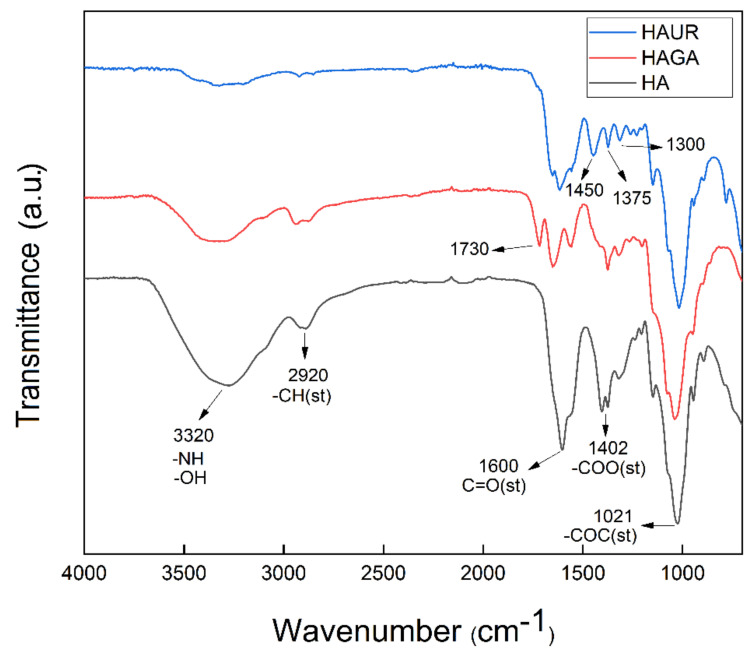
FTIR spectra of microparticles formed from native (HA) and crosslinked (HAGA and HAUR) HA.

**Figure 5 pharmaceutics-13-01878-f005:**
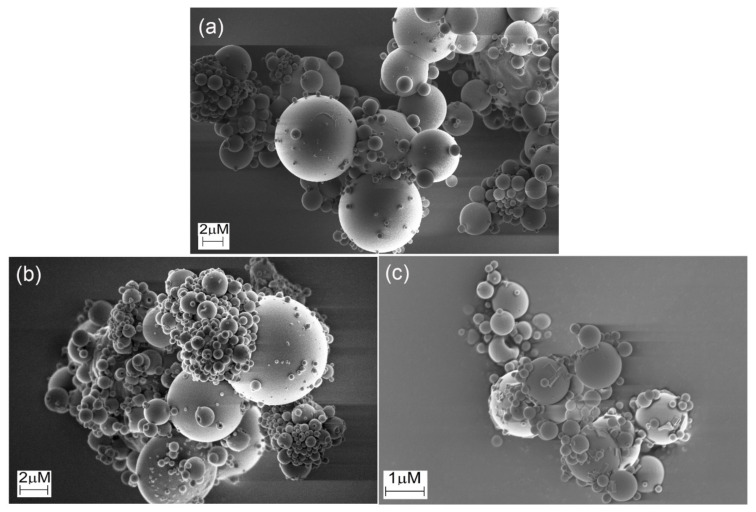
SEM images of microparticles produced by spray drying of (**a**) native HA and HA crosslinked with (**b**) glutaraldehyde (HAGA) and (**c**) urea (HAUR).

**Figure 6 pharmaceutics-13-01878-f006:**
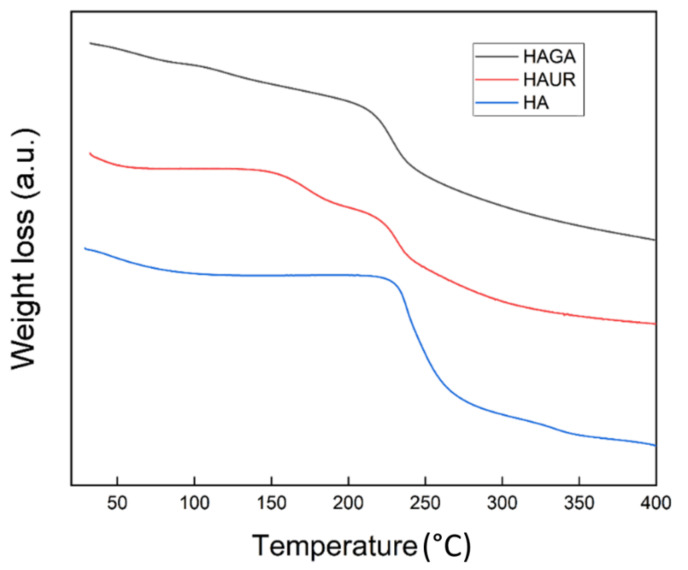
Thermogravimetric analysis (TGA) thermograms of microparticles formed from native (HA) and crosslinked (HAGA and HAUR) HA.

**Figure 7 pharmaceutics-13-01878-f007:**
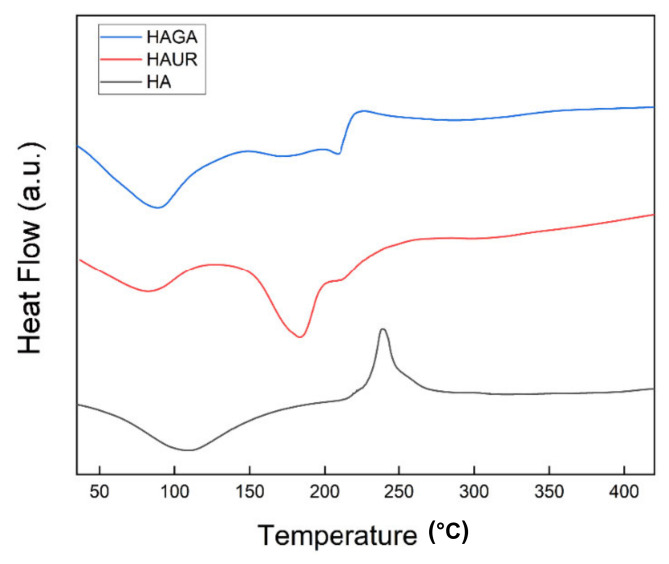
Differential scanning calorimetry (DSC) thermograms of microparticles formed from native (HA) and crosslinked (HAGA and HAUR) HA.

**Figure 8 pharmaceutics-13-01878-f008:**
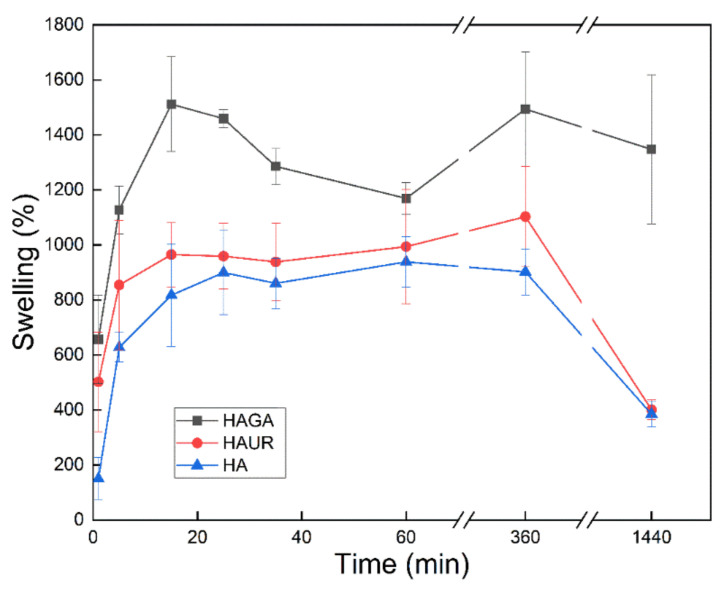
Swelling of microparticles formed from native (HA) and crosslinked (HAGA and HAUR) HA. Bars represent standard error of the mean.

**Figure 9 pharmaceutics-13-01878-f009:**
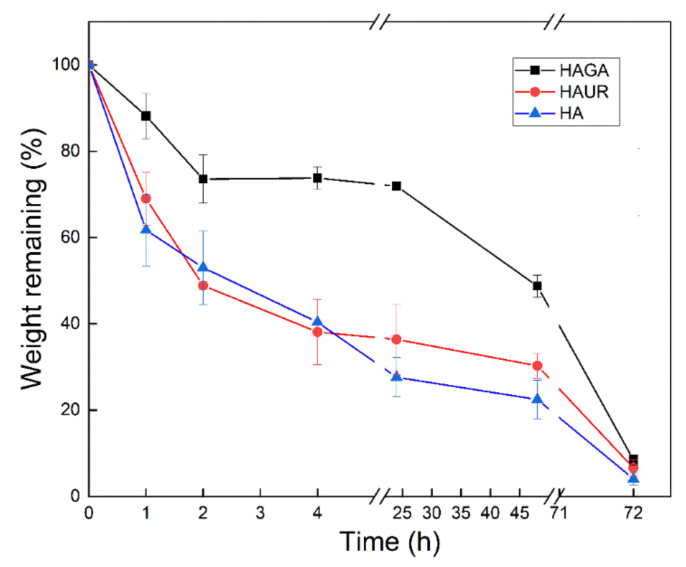
Biodegradation of microparticles formed from native (HA) and crosslinked (HAGA and HAUR) HA. Bars represent standard error of the mean.

**Figure 10 pharmaceutics-13-01878-f010:**
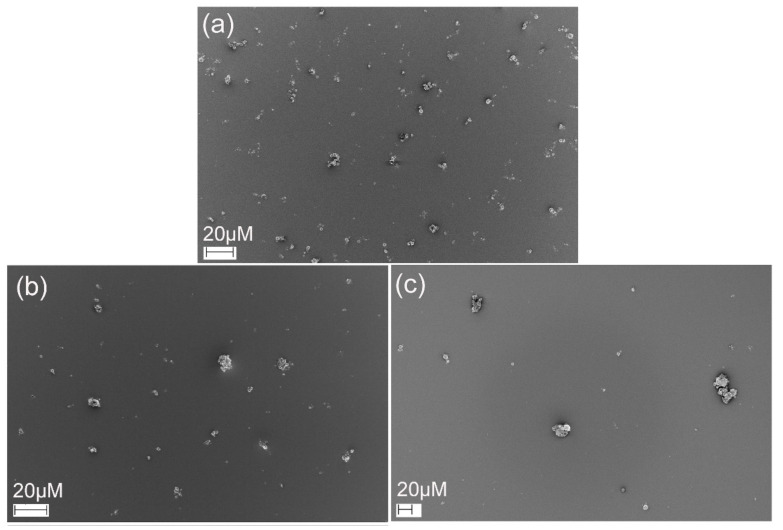
SEM images of HAGA (**a**), HAUR (**b**) and HA (**c**) microparticles deposited using mACI at a 60 L min^−1^ air flow rate.

**Figure 11 pharmaceutics-13-01878-f011:**
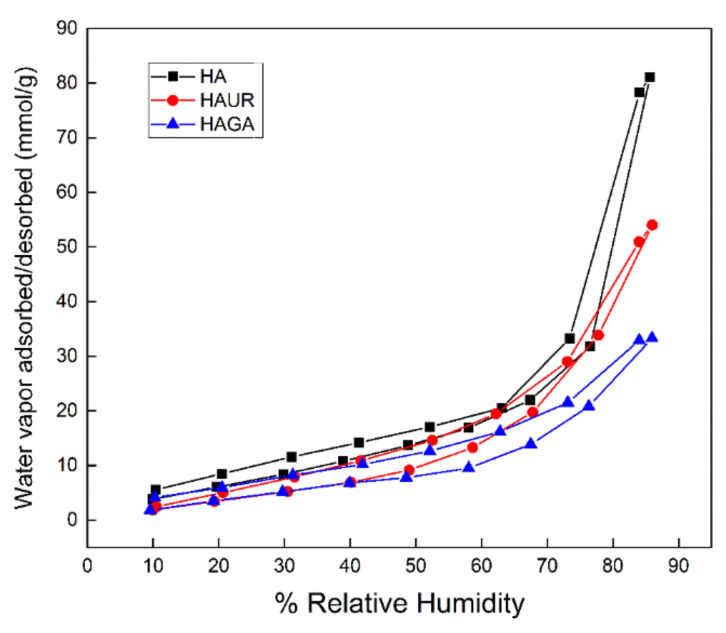
Water sorption/desorption isotherms of HAGA, HAUR, and HA microparticles.

**Table 1 pharmaceutics-13-01878-t001:** Process conditions of spray drying and particle size analysis of HA microparticles along with the crosslinked HA microparticles.

Sample	Inlet Temperature (°C)	Atomizing Air-Flow (L/h)	Drying Air-Flow (L/h)	Feeding Rate (mL/min)	Yield%	Mean (µm)	SD ^a^ (µm)	Median (µm)	Span
HA1	100	375	35	1	28.5	3.2	3.0	2.1	3.5
HA2	110	375	35	1	28.1	2.9	2.7	2.0	2.8
HA3 ^b^	120	375	35	1	22.7	3.0	2.6	2.1	2.2
HA4	120	473	35	1	28.7	2.5	2.5	1.8	2.8
HAUR	120	375	35	1	37.2	2.4	1.1	2.2	1.2
HAGA	120	375	35	1	43.4	2.3	1.1	2.2	1.3

^a^ Pooled standard deviation [30]. ^b^ Also referred to merely as HA.

## Data Availability

All data are available in the main text.

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
