# Peer review of "Hyaluronic Acid Hydrogels for Controlled Pulmonary Drug Delivery—A Particle Engineering Approach"

_pharmaceutics, 2021, doi:10.3390/pharmaceutics13111878_

Round 1

Reviewer 1 Report

This work by Frenning and co-workers used UA and GA to crosslink HA to engineering the final hydrogel for controlled pulmonary drug delivery. The characterization part in this work is nice. The only concern is that the authors lack the real drug delivery experiments in vitro and in vivo.

Author Response

Response to Reviewer 1 Comments

We would like to thank reviewer 1 for giving comments and spending valuable time reviewing our manuscript. Our response to the reviewer 1 comments is provided below.

Point 1: This work by Frenning and co-workers used UA and GA to crosslink HA to engineering the final hydrogel for controlled pulmonary drug delivery. The characterization part in this work is nice. The only concern is that the authors lack the real drug delivery experiments in vitro and in vivo.

Response 1:

We would like to thank the reviewer for the encouraging comment. In this study, which could be considered as the first in a series of two studies, we investigated different formulations based on hyaluronic acid (HA) hydrogel for application in pulmonary drug delivery. The suggested formulations were extensively characterized by several physicochemical methods and the particle engineering aspects of the HA-based hydrogels were evaluated. 

The final goal of this project certainly is to assess in vitro and/or in vivo drug release from the suggested formulations for pulmonary drug delivery. However, the drug release study involves extensive additional investigation of drug dispersion in the formulation (solid dispersion), particle engineering, and finally release monitoring by advanced techniques in the field of pulmonary drug delivery. For this reason, we have decided to address these questions in a second study, as it would be complicated to merge it with the current one.

Reviewer 2 Report

In this manuscript, authors are describing microparticles made from hyaluronic acid, crosslinking both with urea and glutaraldehide.  Microparticles are full characterized and seems to be a potential drug delivery system in lungs. 

Introduction is very well explained, showing the lack of this type of formulations in the market and theit possibilities and advantages. 

In section 2.3.1, authors described "

In this manuscript, authors are describing microparticles made from hyaluronic acid, crosslinking both with urea and glutaraldehide.  Microparticles are full characterized and seems to be a potential drug delivery system in lungs. 

Introduction is very well explained, showing the lack of this type of formulations in the market and theit possibilities and benefits. I just have some curiosities: 

In section 2.3.1, authors described "Due to the different viscosity of the hydrogels and consequently, their nonuniform distribution in the Petri dishes, the obtained dried films were inhomogeneous and brittle". Could authors explain clearly this point? Is the crosslinking homogeneus with the same concentration for all of them? In section 3.5, SEM images demonstrated the non uniformity of the particles. 

In section 3.3., authors indicated a negative charge of the microparticles, could this charge influences in the residence time in the mucosa and not just the size of the particles? 

Conclusions given by the authors are supported by their results.  

Author Response

Response to Reviewer 2 Comments

We would like to thank reviewer 2 for giving comments and spending valuable time reviewing our manuscript. Our response to the reviewer 2 comments is provided below.

Point 1: In this manuscript, authors are describing microparticles made from hyaluronic acid, crosslinking both with urea and glutaraldehide.  Microparticles are full characterized and seems to be a potential drug delivery system in lungs. 

Introduction is very well explained, showing the lack of this type of formulations in the market and theit possibilities and benefits. I just have some curiosities: 

In section 2.3.1, authors described "Due to the different viscosity of the hydrogels and consequently, their nonuniform distribution in the Petri dishes, the obtained dried films were inhomogeneous and brittle". Could authors explain clearly this point? Is the crosslinking homogeneus with the same concentration for all of them? In section 3.5, SEM images demonstrated the non uniformity of the particles. 

Response 1:

An explanation was added in the manuscript (section 2.3.1.) for more clarification. We think that crosslinking was homogenous in the hydrogels as the procedure was implemented under initial stirring and incubation of the mixture for 24h to ensure that the reaction between crosslinker and polymer was completed. The nonuniform structure of the films can be explained by the nonuniform distribution of the hydrogels in the dish (due to high viscosity) and also the drying method.

According to SEM images, the polydispersity of particles is related to the formation of different sizes of the droplets during spray drying, which is normal for this method. The feed suspensions of hydrogels were diluted and dispersed in aqueous media by mechanical stirring before spray drying.

Point 2: In section 3.3., authors indicated a negative charge of the microparticles, could this charge influences in the residence time in the mucosa and not just the size of the particles? 

Conclusions given by the authors are supported by their results

Response 2:

The negative charge may influence the residence time of the particles on the mucosa. The mucoadhesive agents adhere to the mucus layer through the interactions with mucin fibers and physiological turnover of the mucus determines their residence time. It seems that positively charged particles tend to effectively adhere to the negatively charged mucin molecules. However, negatively charged particles tend to diffuse more across the mucus in comparison to positively charged ones. It is of utmost importance that the delivery system cut-off the mucus barrier, avoid rapid clearance, and reach the underlying epithelium. Some other parameters such as ionic strength and pH of the mucus should also be taken into account. (A discussion was added to section 3.3 to clarify the zeta potential in the context of the selected biological application)

As the mucoadhesion of the particles and physiological properties of the lung are not the topic of this study, we honestly prefer to avoid any prediction about the residence time of the particles. For such a claim complementary experiments are needed.

Reviewer 3 Report

The comments for authors are attached. Mainly Figure 1 (high errors regarding chemical formula and codes attribution), Table 1 (inclusion of selected parameters required), discussions on FT-IR ( errors in signals attribution), TGA, DSC (interpretation taken from another published paper based on HAUR -containing excipients- not the case here) and References ([24], [34] to be completed, journals names are >50% not in the abbreviated form- required) must be checked and modified according the journal requirements.  Statistics must be included. Few language/typing errors require the manuscript review/revision.

Author Response

Response to Reviewer 3 Comments

We would like to thank reviewer 3 for giving comments and spending valuable time reviewing our manuscript. Our response to reviewer 3 comments is provided below.

Point 1: The manuscript Hyaluronic acid hydrogels for controlled pulmonary drug delivery ̶ A particle 2 engineering approach, is presenting data on a topical research theme: drug delivery systems for pulmonary use. The novelty is related to the spray-drying method applied to microparticles based on hyaluronic acid. The appropriate preparation and spray drying parameters are established. The characterization part is not at the level of that dedicated to engineering.

Observations:

1. Figure 1, discussions dedicated to FT-IR and TGA/DSC, References chapter must be revised.

a) Figure 1- contains errors regarding the hydrogels formula (i.e. hydrogel cross-linked with GA), and the attribution of codes is wrong (inversed)

Response 1:

We would like to thank the reviewer for noting this error. Figure 1 has been revised and a new version is embedded in the manuscript. Moreover, the discussion and interpretation of FT-IR and TGA/DSC results have been modified when needed (see below). The format of the references has been corrected.

Point 2: b) FT-IR/ In the HAGA spectrum the band at 1730 cm-1 is (with increased probability) from the

aldehyde C=O (unreacted groups); the band at 1600 cm-1 in HA is removed to about 1630cm-1due to superposition with new bands. In the HAUR spectrum the new band at about 1650 cm-1 is usually attributed to amide C=O; lines 371-372, -C-O-C- ether (not ester) band…etc. It normally increased for HAGA comparative to HA. Thus, the attribution of signals must be carefully revised.

Response 2:

We would like to thank you for the valuable corrections. All suggested corrections have been implemented thoroughly in the revised FTIR section.

Point 3: c) The authors sustain that they prepared HAUR hydrogel. Thus, they have not included penthylene glycol – the excipient found in the commercial form (I.R.A. Srl (Istituto Ricerche Applicate Srl, Usmate-Velate, Monza-Brianza, Italy) described in Arianna Fallacara et al., Formulation and Characterization of Native and Crosslinked Hyaluronic Acid Microspheres for Dermal Delivery of Sodium Ascorbyl Phosphate: A Comparative Study, Pharmaceutics 2018, 10, 254; doi:10.3390/pharmaceutics10040254 However, I didn’t considered this aspect as “plagiarism”. Some authors didn’t take attention to the “chemistry part/interpretation”. Abnormal (usually) for N-EU, but considering the pandemic conditions, all is possible.

Response 3:

Our interpretation of the reviewer's comment is that he or she has concerns about the reference to evaporation of pentylene glycol in the interpretation of the TGA and DSC results. Our line of thought during preparation of the original version of the manuscript was that pentylene glycol might be formed during degradation of the urea crosslinks, and hence that the interpretation provided by Arianna Fallacara et al. may remain valid. On second thought, we consider this uncertain, and we have consequently revised the interpretation of the results accordingly. We would like to thank the reviewer for pointing this out and hope that no concerns remain.

Point 4: d) References chapter are not respecting the required form for this journal. Most of journals names are not mentioned with the abbreviated form. Some are written with capital letters. Patent and books are not mentioned in the correct form (i.e [25] and [34]). [25] correct: Citernesi, U.R.; Beretta, L.; Citernesi, L. Crosslinked Hyaluronic Acid, the Process for the Preparation Thereof and Use Thereof in the Aesthetic Field. Patent WO/2015/007773 A1, 22 January 2015

Response 4:

The format of the references has been updated.

Point 5: e) Statistics must be included in the experimental part. SD or SE are presented in Figure 8?

Response 5:

The standard errors of the means are shown in Figures 2, 8 and 9. The definition has been added in the Figure captions.

Point 6: Typing errors are present in the manuscript. Please review the text.

Response 6:

The manuscript has been carefully reviewed as recommended.

Point 7: Some data are missing. Materials: PBS solution concentration, source. Line 160- how was evaluated the thickness of the films. Line 325- units

Response 7:

The concentration of the PBS solution is stated in sections 2.3.8 and 2.3.9 of the revised manuscript, as recommended. The method for evaluation of the thicknesses is now stated and the missing unit has been added.

Point 8: Sentences in the lines 227-228 and 356-358 may be eliminated/avoided

Response 8:

The sentences were deleted from the text as recommended.

Point 9: Table 1. According SEM and DLS data SD for HAGA is higher, not similar to HAUR. The selected data are not included in the table, to compare.

Response 9:

The discussion of particle size was based on the SD and span values, as reported in Table 1. The HA microparticles (HA1-4), presented higher SD and span values. The HAGA and HAUR microparticles had similar SD and span values, which is consistent with DLS data (Figure 3). However, the span for HAGA microparticles is slightly higher than that for HAUR (Table 1).

Point 10: Figure 10. HAUR sample seems to be better dispersed. Check the attribution or chose another microphotograph.

Response 10:

The dispersion of the microparticles could be inferred from the number of microparticles deposited via the mACI and also their tendency to form aggregates, which can easily be seen in SEM images. According to Figure 10, more microparticles were observed for HAGA in comparison to HAUR and HA, which indicates a good dispersion of HAGA microparticles, avoiding adhesion of particles to the walls and/or aggregation during the measurement. Albeit a low number of microparticles are observed for HA and HAUR, the dispersed microparticles also represented aggregates, comprising small particles located on the surface of the larger particles. A new micrograph for Figure 10 (b) was added as recommended.

Point 11: Inclusion of ATI is recommended to complete the study (very high IF)

Response 11:

Unfortunately, we do not understand the meaning of ATI in this context and hence cannot judge the need for its inclusion.

The manuscript may be published only after major revision. It is important for the engineering aspects.

Authors’ response:

The manuscript has been thoroughly revised in accordance with the suggestions by the reviewer.

Reviewer 4 Report

Nikjoo and colleagues described the preparation and characterization of hyaluronic acid particles for pulmonary drug delivery. The study is interesting, well written and particle characterization is very detailed. However, the authors should better correlate their investigations and results with the biological applications and highlight the progress of their study with respect to the state of the art. For this reason, I would suggest MAJOR REVISION.  

  • Lines 52-57: To support the idea that their study is innovative, the Authors should cite more recent references. The most recent reference was published in 2015. 
  • Lines 60-62: Could the Authors be more precise? For instance, they should provide a size range
  • Paragraph 2.3.1: a) The Authors considered hydrogel films with different thickness. Why did they think that the results are not influenced by the thickness of hydrogel films?; b) what does "a determined amount of the film was immersed" mean?
  • Line 141: Please, change 0.7) mm with 0.7 mm)
  • Paragraph 2.3.9: Why did the Authors consider the swollen weights instead of the dried ones?
  • Paragraph 2.3.3: a) Why did the Authors perform the analysis at pH=6?; b) Please, correlate the importance of this analysis with the selected biological application; c ) Line 201: please, check the unit of measure
  • Line 323: Please, rephrase the sentence. Within the reported range, it is not true that "microparticles are smaller than 5 microns"
  • Line 325: Please check "with a flow rate from 375 to 475"
  • Lines 347-351: Please, improve these sentences to strengthen the importance and relevance of the work.
  • Lines 305-306: Why did the Authors agree with the sentence: "It is reported that the cytotoxicity of ... dependent on the concentration"? The presence of inflammation and immune cells could not change the microenvironment? Please, discuss on this point.
  • Paragraph 3.3: Which is the importance of zeta potential in the context of the selected biological application? Why did the Authors perform this analysis?
  • Lines 482-484: What did "decomposition" mean? Please, rephrase the sentence.
  • Lines 510-512: I would suggest to rephrase this sentence, not to diminish the importance of the results. Please, add a reference explaining how the environment could be improved
  • Minor comment: please, change in-vitro, in-vivo with in vitro, in vivo

Author Response

Response to Reviewer 4 Comments

We would like to thank reviewer 4 for giving comments and spending valuable time reviewing our manuscript. Our response to reviewer 4 comments is provided below.

Point 1: Nikjoo and colleagues described the preparation and characterization of hyaluronic acid particles for pulmonary drug delivery. The study is interesting, well written and particle characterization is very detailed. However, the authors should better correlate their investigations and results with the biological applications and highlight the progress of their study with respect to the state of the art. For this reason, I would suggest MAJOR REVISION.  

Lines 52-57: To support the idea that their study is innovative, the Authors should cite more recent references. The most recent reference was published in 2015. 

Response 1:

This is a good point raised by the reviewer. The following recently published papers were cited to support the innovative nature of the study.

Line 52-57:

[8] Buhecha, M.D.; Lansley, A.B.; Somavarapu, S.; Pannala, A.S. Development and Characterization of PLA Nanoparticles for Pulmonary Drug Delivery: Co-Encapsulation of Theophylline and Budesonide, a Hydrophilic and Lipophilic Drug. J. Drug Deliv. Sci. Technol. 2019, 53, 101128.

[9] Athamneh, T.; Amin, A.; Benke, E.; Ambrus, R.; Leopold, C.S.; Gurikov, P.; Smirnova, I. Alginate and Hybrid Alginate-Hyaluronic Acid Aerogel Microspheres as Potential Carrier for Pulmonary Drug Delivery. J. Supercritical Fluids 2019, 150, 49-55.

[10] Fernández-Paz, E.; Feijoo-Siota, L.; Gaspar, M.M.; Csaba, N.; Remuñán-López, C.  Microencapsulated Chitosan-Based Nanocapsules: A New Platform for Pulmonary Gene Delivery. Pharmaceutics 2021, 13, 1377.

Line 557:

[59] Nishimura, S.; Takami, T.; Murakami, Y. Porous PLGA Microparticles Formed by “one-Step” Emulsification for Pulmonary Drug Delivery: The Surface Morphology and the Aerodynamic Properties. Colloids Surf. B Biointerfaces 2017, 159, 318-326.

Point 2: Lines 60-62: Could the Authors be more precise? For instance, they should provide a size range

Response 2:

This is a good suggestion. The following elaboration has been added to the text:

For example, an increase in the size of swellable biocompatible microparticles has been reported in PBS, where the size of different dry formulations increased from 3–3.5 μm to 30.2–38.1 and 82.7–91.0 μm within 6 and 20 min, respectively [11].

[11] El-Sherbiny, I.M.; Smyth, H.D. Controlled Release Pulmonary Administration of Curcumin using Swellable Biocompatible Microparticles. Mol. Pharm. 2012, 9, 269-280.

Point 3: Paragraph 2.3.1: a) The Authors considered hydrogel films with different thickness. Why did they think that the results are not influenced by the thickness of hydrogel films?; b) what does "a determined amount of the film was immersed" mean?

Response 3:

a) The thickness of the hydrogel films does influence the swelling results. Thus, an identical amount of hydrogels was poured into the Petri dish for each hydrogel type (HAGA and HAUR) and for each type, a similar thickness was observed (around 481±58 µm for the HAGA films, i.e. HAGA1-HAGA16, and 360±46 µm for the HAUR films, i.e. HUR1-HAUR20). The swelling results were primarily compared and discussed for each hydrogel type.

b) As suggested by the reviewer, the amount used is reported in the revised version (section 2.3.1).

Point 4: Line 141: Please, change 0.7) mm with 0.7 mm)

Response 4:

This writing error has been corrected.

Point 5: Paragraph 2.3.9: Why did the Authors consider the swollen weights instead of the dried ones?

Response 5:

In-vitro biodegradation was recorded for 72h at different time intervals. The weight of the (swollen) microparticles was compared with the initial mass of the (swollen) particles to calculate the remaining weight (%) at each interval. To use a similar analysis using dried weights, the swollen microparticles must be dried and weighed in each interval and again should swell to continue the measurement. It would make the analysis more complicated and time-consuming, thus the swollen weight of the microparticles has been used as a standard method for biodegradation analysis and the remaining weight (%) was reported. This is a standard method that previously has been used to assess biodegradation of microparticles (see refs 44, 45, and 55).

Point 6: Paragraph 2.3.3: a) Why did the Authors perform the analysis at pH=6?; b) Please, correlate the importance of this analysis with the selected biological application; c ) Line 201: please, check the unit of measure

Response 6:

a) The method was updated in section 2.3.3.; the measurement was implemented in NaCl electrolyte (10 mM) at the temperature of 25 °C.

b) In section 2.3.3., we presented the method that has been used for measuring zeta potential. However, a discussion was added to section 3.3., to correlate the importance of this analysis with the selected biological application as recommended (reviewer suggested similar comments for section 3.3).

c) Prefixes have been removed and SI units are used throughout.

Point 7: Line 323: Please, rephrase the sentence. Within the reported range, it is not true that "microparticles are smaller than 5 microns"

Response 7:

The sentence was rephrased.

Point 8: Line 325: Please check "with a flow rate from 375 to 475"

Response 8:

The missing units have been inserted.

Point 9: Lines 347-351: Please, improve these sentences to strengthen the importance and relevance of the work.

Response 9:

The indicated lines have been rewritten in order to better convey our message:

It is worth mentioning that the obtained particle sizes and size distributions are in line with prior results reported for spray-dried HA powder formulations for pulmonary drug delivery [12,23,34,36]. Spray drying of hydrogels is challenging, however, the narrow size distribution (span=1.2 and 1.3) and mean diameters around 2.2 μm for the crosslinked HA confirm the possibility of using spray drying for preparation of inhalable dry powders based on hydrogels. The results indicate that spray drying not only is convenient for the manufacturing of inhalation powders from HA suspensions but also for the preparation of particles from crosslinked structures.

Point 10: Lines 305-306: Why did the Authors agree with the sentence: "It is reported that the cytotoxicity of ... dependent on the concentration"? The presence of inflammation and immune cells could not change the microenvironment? Please, discuss on this point.

Response 10:

The claim is based on the literature, however, an extensive purification method was used to purify synthesized hydrogels, eliminating unreacted and toxic agents as reported in section 2.2. Moreover, a brief discussion has been included in the section.

Point 11: Paragraph 3.3: Which is the importance of zeta potential in the context of the selected biological application? Why did the Authors perform this analysis?

Response 11:

We perform the zeta potential analysis to understand and explain any interaction between different microparticles in suspension. This is important to know it during characterization (e.g. DLS) as well as for future drug release analysis. Accordingly, the relatively high negative values of the zeta potential indicated a stable colloidal system for the microparticles.

The following discussion has been added in section 3.3., to highlight the importance of the zeta potential in the context of the selected biological application:

“Further, the surface charge of the microparticles influences the mucoadhesive, mucopenetration, and cell uptake characteristics in mucosal drug delivery. According to a previous study, positively charged polymeric particles represented mucoadhesive properties due to the interaction with negatively charged regions of mucus. Where, the negatively charged particles have proved to enhance mucopenetration through minimizing interaction with the negatively charged parts of mucus, as well as cellular uptake via minimizing interactions with positively charged groups of the lipid membrane [39].

The application of HA as a mucoadhesive biopolymer was reported earlier [40, 41]. It was also approved that HA is capable of enhancing the penetration of drugs, comparable to chitosan derivatives [42]. Conclusively, in the mucosal drug delivery system, the surface charge plays a key role in predicting drug uptake capacity even though it is not the only effective parameter.”

Point 12: Lines 482-484: What did "decomposition" mean? Please, rephrase the sentence.

Response 12:

Decomposition means disintegration in this sentence. The sentence was rephrased as follows:

The disintegration of the HAUR implies that swelling is not the only parameter to measure the stiffness of the hydrogels and other factors such as degree of crosslinking and stability of the crosslinked bonds are of utmost importance.

Point 13: Lines 510-512: I would suggest to rephrase this sentence, not to diminish the importance of the results. Please, add a reference explaining how the environment could be improved

Response 13:

The sentence was rephrased and an explanation was added as follow:

However, the microenvirnment in the lung (e.g pulmonary surfactants) may also influence the in vivo biodegradation rate. The inclusion of Pluronic F-108 surfactant in a chitin-based formulation has been reported for localized administration in lung solid tumors. Accordingly, the existence of F-108 surfactant in the formulation accelerated the lysozyme-induced biodegradation and consequently the drug release from the microparticles [57].

Point 14: Minor comment: please, change in-vitro, in-vivo with in vitroin vivo

Response 14:

The correction was done throughout the manuscript.

Round 2

Reviewer 3 Report

There is still an error in the figure 1. Replace N from the HAGA (MPs) formula with (CH2)3.

Reviewer 4 Report

I would like to thank the Authors for providing a new version of their manuscript. In the new version, they replied to most of my comments. For this reason, I would suggest to accept the manuscript in the present form.